# Approaches and Technologies in Male Fertility Preservation

**DOI:** 10.3390/ijms21155471

**Published:** 2020-07-31

**Authors:** Mahmoud Huleihel, Eitan Lunenfeld

**Affiliations:** 1The Shraga Segal Department of Microbiology, Immunology and Genetics, Faculty of Health Sciences, Ben-Gurion University of the Negev, Beer Sheva 84105, Israel; 2The Center of Advanced Research and Education in Reproduction (CARER), Ben-Gurion University of the Negev, Beer Sheva 84105, Israel; lunenfld@bgu.ac.il; 3Department of OB/GYN, Soroka Medical Center, Beer Sheva 8410501, Israel; 4Faculty of Health Sciences, Ben-Gurion University of the Negev, Beer Sheva 84105, Israel

**Keywords:** male fertility preservation, male infertility, chemotherapy, in vitro culture of spermatogonial cells, testis, spermatogenesis, organ culture, three-dimension in vitro culture system, cancer and male infertility, cytokines and male infertility

## Abstract

Male fertility preservation is required when treatment with an aggressive chemo-/-radiotherapy, which may lead to irreversible sterility. Due to new and efficient protocols of cancer treatments, surviving rates are more than 80%. Thus, these patients are looking forward to family life and fathering their own biological children after treatments. Whereas adult men can cryopreserve their sperm for future use in assistance reproductive technologies (ART), this is not an option in prepubertal boys who cannot produce sperm at this age. In this review, we summarize the different technologies for male fertility preservation with emphasize on prepubertal, which have already been examined and/or demonstrated in vivo and/or in vitro using animal models and, in some cases, using human tissues. We discuss the limitation of these technologies for use in human fertility preservation. This update review can assist physicians and patients who are scheduled for aggressive chemo-/radiotherapy, specifically prepubertal males and their parents who need to know about the risks of the treatment on their future fertility and the possible present option of fertility preservation.

## 1. Spermatogenesis

Spermatogenesis is the process of spermatogonial stem cell (SSC) proliferation and differentiation through meiotic stages to generate sperm [1,2,3,4]. Endocrine and paracrine systems are involved in spermatogenesis regulation [1,2,3,4,5,6]. The main endocrine factors are the GnRH and gonadotropic hormones FSH and LH. FSH directly and specifically affects Sertoli cells in order to produce different factors, such as androgen binding proteins (ABP), transferrin, inhibin, androgen receptor, while LH specifically affects Leydig cells to produce testosterone. Because germ cells do not express receptors for FSH and LH, their effects are indirect on germ cells through various factors/substances that are secreted by Sertoli, Leydig, and other affected cells in the testes [1,2,3,4,5,6].

Adult stem cells in mammals have the capacity of self-renewal and the production of differentiated daughter cells (they can replicate indefinitely). The progenitor cells are an intermediate cell population between stem and differentiated cells. These cells have only the capacity of self-renewal and differentiation (they can divide only a limited number of times) and, thus, play a homeostatic role in maintaining the development of complete spermatogenesis [1,2,3,4,5,6].

In rodents, spermatogonial cells are grouped into type A, whereas in the mouse, seven types of A spermatogonia have been described [A (single), A (pair), A (aligned), A1–A4] when A (single) cells are considered to be the SSCs. However, in human, spermatogonial cells are grouped into Adark and Apale, when Adark is considered as SSCs. The SSCs have the capacity of self-renewing with a low rate of proliferation when compared to their progeny of spermatogonial cells (Apair and Aaligned in the mouse and Apale in human) that have the capacity to self-renew, but with a high rate of proliferation [1,2,3,4,5,6].

## 2. Cancer, Chemotherapy, and Male Infertility

### 2.1. Cancer and Male Infertility

It was shown that different types of cancers, including leukemia, lymphoma, testicular cancer, non-Hodgkin’s disease, gastrointestinal malignancy, and musculoskeletal malignancy, affect sperm parameters and the quality of thawed cryopreserved sperm [7,8]. Recently, using a mature animal model of leukemia, we showed that acute myeloid leukemia (AML) significantly decreased sperm parameters, increased spontaneous acrosome reaction, and decreased both male fertility capacity and number of offspring [9]. Thus, cancer may lead to male subfertility/infertility.

### 2.2. Cancer Patients and Chemotherapy

In recent years, the number of young male cancer survivors has increased drastically (>75%). This is mainly due to early diagnosis and improved cancer treatment protocols [10]. Therefore, the issue of infertility treatment to maintain the ability to genetically father one’s own children is a major concern for those young survivors who were treated with gonadotoxic agents [4,11,12].

Fertility is often impaired after chemotherapy and radiation therapy and may lead to azoospermia [4,12]. Cryopreservation of semen before the commencement of cancer treatment is currently the only method of preserving future male fertility. Obviously, this technique is not an option for prepubertal male cancer patients, since they do not yet generate sperm. Until now, fertility preservation for these patients has not been an option [4,10].

### 2.3. Gonadotoxic Agents and Male Infertility

Cells of the seminiferous tubules (dividing spermatogonial cells) are the most sensitive to damage that is caused by gonadotoxic agents, such as chemotherapy and radiotherapy. Restoration of spermatogenesis depends on the type, dose, and duration of treatment of the drugs and radiation used. As a result, young patients cured of cancer often suffer a prolonged and/or permanent decrease in sperm parameters or even azoospermia and are infertile [4,13,14]. The most sensitive testicular cells to chemotherapeutic drugs and irradiation are those that undergo constant proliferation and differentiation into spermatogonia [15,16,17,18]. The death/depletion of these cells results in loss of the following and developed stages of germ cells and, thus, to a decrease/loss in sperm counts. Therefore, the restoration of spermatogenesis and fertility depends on the survival of active SSCs and their ability to differentiate into spermatozoa [15].

Rapid restoration of complete spermatogenesis following gonadotoxic treatment depends on the presence of active SSCs and their microenvironment (somatic cell compartment) [17]. However, if some of these cells are destroyed, as occurs with irradiation and some chemotherapeutic agents, recovery is more gradual [19,20]. In mice, it was shown that the recovery of spermatogenesis is directly proportional to the number of SSCs destroyed. It was also demonstrated that gonadotoxic agents could damage somatic cells in the testis. In rat testes, no recovery of the seminiferous epithelium was demonstrated, even though numerous SSCs survived cytotoxic treatment [21,22].

If the damage is severe (as a result of high doses of chemo-/radiotherapy and/or a combination of these treatments), all of the SSCs commit to apoptosis or, alternatively, damaged Sertoli cells are unable to support the maintenance, growth, and development of SSCs. These events may lead to the complete depletion of the pool of SSCs in the seminiferous tubules, resulting in permanent sterility [14,23].

### 2.4. Recovery of Human Spermatogenesis after Cancer Therapy

The restoration of sperm development in adults following gonadotoxic treatments depends on the presence of quiescent or biologically active SSCs (Adark) that mitotically divide to generate a progeny of more differentiating cells (Apale). As has been mentioned, the possibility of spermatogenesis recovery is affected by the dose and type of gonadotoxic therapy used and individual response [20]. Up to now, studies in primates to develop strategies to protect gonads from these therapies have failed [4,14,23].

## 3. Options for Male Fertility Preservation

Very few and limited options are available for bypassing/minimizing the loss of fertility in male cancer patients following treatment with chemo-/radiotherapy, and most of the suggested options are still under study (Table 1 and Figure 1).

### 3.1. Cryopreservation of Sperm

Cryopreservation of sperm has been extensively used for post pubertal adolescents and adults, and success rates have dramatically improved due to the introduction of intra-cytoplasmic sperm injection (ICSI) combined with mature oocyte retrieval. Unfortunately, only about 24% of men in this age group have cryopreserved semen prior to oncologic treatment [24,25]. In some of the azoospermic patients, sperm can be recovered surgically from small focal areas of spermatogenesis in the testes using testicular sperm extraction (TESE) methods. The freshly isolated testicular sperm can be used to fertilize oocytes by ICSI or freezing them for future ICSI.

It was previously shown that the live birth rate (using assistance reproductive technologies (ART)) was similar between cancer and non-cancer male infertility patients. These results indicate the value of sperm cryopreservation before starting cancer treatments, and the importance for oncologists to consult with young cancer patients to cryopreserve sperm prior to starting these treatments [26].

The main limitation of this technology in humans is that these techniques are only applicable for adult patients who can generate mature sperm cells, and not for prepubertal or other males who cannot produce mature sperm. Furthermore, sperm quantity is limited and quality is impaired.

### 3.2. Cryopreservation of Human Testicular Tissue

Because prepubertal boys cannot generate sperm, a potential alternative approach for preserving their fertility involves cryopreservation of their testicular tissue. This approach is also appropriate for adolescent and adult patients who may already be found to be azoospermic at the time of cancer diagnosis or post chemo-/radiotherapy [4,14,25]. Testicular cells can be cryopreserved as a cell suspension or in the form of tissue. The optimal method of storage is still unclear due to limited research that compared between cryopreserved testicular tissue and cell suspension. However, cryopreservation of testicular tissue provides the future option to use it for both tissue (such as testicular grafting or organ culture) or cell therapy (such as germ cell transplantation and in-vitro maturation), whereas the cryopreservation of testicular cell suspension can be used in the future only for cell therapy. In addition, cryopreservation of testicular tissue will maintain the microenvironment niche of the SSCs (mainly the somatic cells, Sertoli, Leydig, and peritubular cells), which may increase the viability and functionality of the SSCs following tissue thawing when compared to thawed cryopreserved cells.

The cryopreserved testicular cells or tissue could be used for prepubertal human male fertility preservation with different approaches that are already used in rodents [27,28,29,30]. The main limitation of using autologous testicular tissue or cells in human therapy from cancer patients is the possibility of the presence of residual cancer cells, which may restore and evoke the disease.

The survival of spermatogonia has been demonstrated after immature testicular tissue (ITT) freezing, which is considered to be ethically acceptable. The majority of cryopreserved samples showed reproductive potential [31]. Additionally, testicular growth of the biopsied testis was not impeded in comparison to the non-biopsied contralateral testis up until one year after surgery [32]. Cancer incidence was not increased, and the long-term survival rate was not altered after the transplantation of in-vitro propagated murine spermatogonial stem cells (SSCs) in busulfan-treated recipients as compared to non-transplanted busulfan-treated controls [33].

### 3.3. Germ Cell Transplantation

This technology was developed by Brinster et al. in 1994 [34,35]. It includes the injection (transplantation) of isolated testicular germ cells from lacZ transgenic mice into the seminiferous tubules of infertile mice. After a few months, spermatogenesis developed, and the mice became fertile. Additionally, their progeny contained the lac Z gene [34]. This technology was also successful using other species, such as rats, goats, sheep, dogs, pigs, and monkeys of different ages, and after different periods of germ cell cryopreservation [34,35,36,37,38,39,40,41,42,43,44,45,46,47,48,49,50,51,52]. The recovery of spermatogenesis in mice using SSC transplantation was demonstrated even after 14 years post cryopreservation [27,53]. This may suggest that SSCs remain biologically active, even after a long period of cryopreservation. Recently, the restoration of functional sperm production in irradiated pubertal rhesus monkeys by spermatogonial stem cell transplantation was demonstrated even though the success rate of this procedure was low [54]. In humans, it was reported that cryopreserved testicular cells from non-Hodgkin’s lymphoma patients were transplanted into their testes after recovery from cancer [55,56]; however, follow-up for these patients was not reported. Xeno-germ cell transplantation of baboons, marmosets, and humans into mice only colonized, but did not differentiate [28]. In addition, a limitation of xeno-germ cell transplantation is the risk of possible retrovirus transmission to humans if the developed cells are used for fertility therapy. The general limitations of this technology include the need for millions of testicular cells in order to contain enough SSCs to restore spermatogenesis. Nagano et al., suggested that transplantation of one-million of mouse testicular cells induced spermatogenesis by 19 SSCs [57]. It is well known that the number of SSCs in the testis is very low. In adult mouse testes, these are around 0.03% [58], and spermatogonial cells comprised about 3% of the cells in testis of prepubertal boys [59]. Therefore, the development of efficient methods to isolate/enrich SSCs and increase their numbers (proliferation) in vitro are crucial to the successful use of this technology. In addition, the efficiency of SSC transplantation is very low and needed several months (more than eight months in busulfan treated adult mice and five months in W adult mice) to restore fertility [35,60,61]. However, when the recipients were immature W mice, the efficiency was significantly improved, and fertility was demonstrated after three months [61].

The main limitation of this technology in humans is the possible contamination of testicular cells with cancer cells that may reintroduce malignancy to the patient after recovery. Today, there is no accurate and safe method that isolates pure SSCs from the testis of patients [50,62,63], in addition to the very small amount of testicular biopsy that could be used, it also contains very small numbers of these cells [4,14,24,27,29,58,64,65].

### 3.4. Testicular Tissue Autograft/Xenograft

The use of testicular tissue to induce spermatogonial cells of the donor to complete spermatogenesis in vivo or in vitro (organ culture) is based on the availability of the somatic cells to permit proper SSC development. This microenvironment that provides optimal cell–cell interactions and the spatial environment for complete spermatogenesis is produced by the seminiferous tubules and interstitial compartment. This technology was proven successful in both autografting/xenografting and in vitro as organ culture [27,28,29,30]. Testicular tissue xenograft was successfully performed in immunodeficient mice while using testicular tissue from different species, which was grafted under the skin, producing complete spermatogenesis. Xenografting of testis tissue from immature males of different mammalian species, such as hamsters, goats, monkeys, bulls, pigs, and cats, into immunodeficient mice also resulted in complete spermatogenesis [4,14,64,66]. Likewise, live offspring were reported from sperm generated from testicular grafts of immature mice [67] and fertilization-competent sperm from testicular xenografts of immature rhesus monkeys into mice [37]. It was reported that the autotransplantation of fresh testicular tissue from prepubertal marmosets into the scrotum led to the development of full spermatogenesis [68]. Recently, autologous grafting of cryopreserved prepubertal rhesus testis under the back skin or scrotal skin of castrated pubertal rhesus macaques matured to produce functional sperm and offspring [25]. Unfortunately, experiments involving the xenografting of human prepubertal cryopreserved testicular tissue into immunodeficient mice has not yet demonstrated the development of spermatid differentiation [25,27,69,70,71,72]. It was shown that successful spermatogenesis in testicular tissue xenografts are affected by development through spermatogenesis, in addition to the timing of development and efficiency [64]. An additional factor that affects testes’ xenograft survival and development is the age or developmental stage of the donor [64].

It should be noted that testicular xenografts in mice may lead to the transmission of viruses to the human germ line. Furthermore, testicular autografts in human cancer patients have the limitation of possibly reseeding malignant cells, promoting cancer recurrence. However, this technology might be considered in non-cancer patients.

### 3.5. Testicular Cell Xenograft

When compared to testicular tissue xenografting, this technique may allow for more exposure of testicular cells to the new environment. Structural organization of the seminiferous tubules might influence the development of xenograft testicular tissue [4,14,64,66]. Xenotransplantation of testicular cells (a combination of somatic and germ cells in different percentages) from neonate porcine (under the skin of nude mice) led to the development of complete spermatogenesis (after 30 weeks), including the generation of sperm (in 10% of the formed seminiferous tubules) [73]. In another study, ectopically grafting testicular cells from embryonic or neonatal mice and rats to mice led to the development of round spermatids. Using ICSI technology, these cells were able to develop embryos, resulting in the birth of normal pups [74]. In addition, the xenotransplantation of a heterogeneous cell suspension of sexually immature lambs (two weeks old) under the dorsal skin of nude mice led to the development of complete spermatogenesis after 40 weeks [75]. The transplantation of testicular cells from prepubertal rhesus monkeys into the testes of irradiated adult monkeys formed seminiferous tubules with full spermatogenesis nine months post transplantation [76]. To the best of our knowledge, this system has not yet been performed in humans.

### 3.6. Organ Culture

As mentioned above, the main benefit of using testicular fragments to develop spermatogenesis in vitro is the presence of testicular somatic cells and the three-dimension (3D) microenvironment that is suitable for the optimal development of SSCs to complete spermatogenesis [25,26,27,28,29,77]. Spermatogenesis development using organ culture to proceed to the meiotic stages was performed in the past without success [77,78,79,80]. However, Sato et al. recently succeeded in inducing the development of fertile sperm using an in-vitro culture of small fragments (around 3 mm) of testicular tissue from immature mice [81]. This system led to the development of flagellated spermatozoa from cryopreserved testicular tissues of immature mice [80,81]. It was shown that, for the effective development of human spermatogonial cells, it is suggested to grow the testicular human organ at 34 °C and in the presence of gonadotropins, which may increase the long-term period of Sertoli cell survival [82,83,84,85].

The successful application of this system using human testicular tissue could circumvent testicular auto-transplantation to generate sperm and the possibility of reintroducing cancer cells to cured patients. However, neither fertilization nor implantation of human tissue has been reported.

### 3.7. In-Vitro Cultures of SSCs

#### 3.7.1. Two- and Three-Dimensional Culture Systems

In-vitro development of complete spermatogenesis from spermatogonial cells may overcome the limitations of restoration of cancer using organ culture or germ cell transplantation from cancer patients, and possible retrovirus transmission by using xenotransplantation techniques. In addition, the induction of proliferation of spermatogonial cells in vitro may promote the use of autologous human germ cell transplantation.

Several in-vitro culture systems have been used to induce proliferation and differentiation of SSCs from different species to the meiotic and post-meiotic stages. These systems included the addition of adherent cells, conditioned media, or recombinant growth factors while using different matrices, such as collagen, laminin, and others, to mimic seminiferous tubules in in-vivo conditions [1,77,86,87]. However, these systems could mainly induce the proliferation of SSCs and not complete the spermatogenesis process. The induction of proliferation of human SSCs isolated from both normal men and prepubertal cancer and azoospermic patients using an in-vitro culture system was reported [88,89]. To date, factors that are capable of inducing in-vitro differentiation of SSCs from any species have not yet been defined. Our group was the first to suggest the use of a methylcellulose culture system (MCS) and soft-agar-culture system (SACS) as possible three-dimensional (3D) matrices to grow and develop spermatogonial cells in vitro [87,88,89,90,91,92]. Using these two novel 3D culture systems (MCS and SACS), which are more representative of in-vivo conditions, we could induce the proliferation and differentiation of spermatogonial cells from normal and busulfan-treated immature mice to the meiotic and post-meiotic stages, and even the generation of sperm-like cells (elongated spermatid with head, neck, and tail) [87,90,92,93]. The development of meiotic and post-meiotic stages was examined after 3–6 weeks of culture. However, the efficiency of this system to generate sperm-like cells was very low, in addition to the inability to examine the capacity of their fertility. We also demonstrated the capacity to induce the proliferation and differentiation of spermatogonial cells from prepubertal monkeys to meiotic (crem-1 positive cells) and post-meiotic (acrosin positive cells) stages in vitro using MCS [91]. Recently, we showed the induction of the proliferation of spermatogonial cells from biopsies without sperm of azoospermic patients and their differentiation into meiotic cells in MCS [94,95]. We also demonstrated the presence of spermatogonial cells in testicular biopsies of prepubertal cancer patients before aggressive chemotherapy and their differentiation in MCS to meiotic and post-meiotic cells and, in one case, the generation of sperm-like cells [96,97]. We suggest that 3D culture systems may provide the SSCs with a spatial and microenvironment similar to those present in the seminiferous tubules and crucial for the development of spermatogenesis such as the presence of testicular somatic cells and the 3D tubular structure. This 3D in-vitro culture system still requires optimization in order to efficiently generate sperm.

#### 3.7.2. 3D Bioprinted Scaffold

Although in-vitro spermatogenesis has mainly been achieved (not efficiently) in rodent organ culture and/or 3D in-vitro culture systems [27,28,29,30,31,90,91,92,93], the optimal conditions that provide an optimal 3D spatial environment with similar cellular composition of the seminiferous tubule to efficiently induce the development of spermatogonial cells in order to complete maturation and generation sperm was not yet published. A new culture system that provides alginate-based hydrogel and 3D bioprinting was developed to control scaffold design and cell deposition in order to preserve testicular cells in their native 3D spatial and cellular microenvironment to induce complete in-vitro spermatogenesis. Recently, the first report of in-vitro spermatogenesis in mouse testicular constructs generated by culturing single cell suspensions on 3D bioprinted cell-laden scaffolds and cell-free scaffolds was published [98]. Cell spheres were generated, but tubule-like structures did not develop. However, patches of differentiated post meiotic cells, including round and elongated spermatids, have been demonstrated [98]. The fertility capacity of the developed spermatids was not examined. This system still needs to be optimized at the different levels of the scaffold and cultured cells for providing tubule-like structures and more efficient spermatid development.

#### 3.7.3. Testicular Organoids

The term organoid is used to describe 3D structures (up to two millimeters) that are composed of aggregates of cells that reorganize after cell dissociation of specific tissue [99,100,101]. These new structures show some histology and biological activity similar to the original tissue [99]. The development and functionality of these organoids depends on optimal conditioned media, nutrients, and oxygen provided in vitro, since they do not form the vasculature system [99,100,101,102]. It is possible that the development/formation of these organoids may assist researchers in understanding the cell–cell interactions and functionality of the original tissue. The development of functional organoids without the formation of testis-like histology from adult human SSCs was recently demonstrated [103]. They were co-cultured with Leydig and Sertoli cells in the presence of extracellular matrix (ECM) from human testis and led to the development of haploid cells after three weeks of culture [103]. Another study showed the development of active human testicular organoids after four weeks of culture (without similarity to the histology of the testis). These organoids did produce testosterone and inhibin B, but did not induce differentiation of the proliferative spermatogonial cells [104]. Additionally, culturing testicular cells from immature rats in a three-layer gradient system led to the development of organoids (composed mainly of Sertoli and germ cells) with tubule-like structures after seven days of culture. These structures showed the development of a functional testicular blood barrier between the Sertoli cells without differentiation of the spermatogonial cells [105]. An additional study showed the development of mouse spermatogonial cells to meiotic and post meiotic cells in organoids when cultured in fabricated testis-derived scaffolds [106].

#### 3.7.4. Microfluid System and Organ-on-Chip Technology

The lack of circulatory system is one of the main limitations of 3D in-vitro systems. Therefore, organ-on-chip in a microfluid device may overcome this limitation and provide a dynamic condition of nutrient and gas circulation mimicking the in-vivo conditions. This system enabled the use of small amounts of media and control of their composition, diffusion, and temperatures very close to the cells/tissue present in the device. Thus, the use of the 3D in-vitro culture system may enable the capacity of spermatogonial cells to complete/efficient maturation and the generation of sperm in testicular tissue or isolated tubular cells cultured in such a system. Recently, the development of organ-on-chip of neonate mouse testicular tissue was reported [107,108,109]. In this system, a continuous perfusion of media was supplemented to the design chambers through vasculature-like structures. The media arrived to the testicular tissues only by diffusion. The testicular tissue showed efficient spermatogenesis for around six months, with the ability to develop round spermatids with fertility capacity, as demonstrated by the generation of healthy offspring following spermatid injection to oocytes [107]. Recently, our group developed a mouse microfluid system with two chambers that contained isolated cells from seven-day-old mice in methylcellulose. These chambers were provided by continuous diffused conditioned media similar to MCS. Our preliminary and unpublished data show the development of organoids with seminiferous tubule-like structures (with a histology of seminiferous tubules that contain peritubular cells and active Sertoli cells) after 5–7 weeks of culture, as examined by confocal microscope. These organoids contained, in addition to premeiotic cells (VASA and PLZF positive stained cells), developed meiotic (CREM positive stained cells) and post meiotic cells (ACROSIN positive stained cells), as examined by confocal and immunofluorescence microscopes. This microfluid system showed a significant increase in cell viability and the percentages of developed 1N cells when compared to MCS, as examined by trypan blue staining and FACS analysis, respectively.

The success of these systems in humans, including sperm generation, may overcome limitations, such as the small testicular biopsies used with a very low number of SSCs and the concerns of the possible restoration of cancer cells to cured patients. This may benefit not only prepubertal cancer patients, but also non-obstructive azoospermic patients who do not have sperm in their biopsies, but in whom spermatogonial cells are present. On the other hand, the capacity of fertility of the generated sperm in this system still needs to be confirmed, in addition to genetic and epigenetic stability, as in all in-vitro systems.

### 3.8. Induced Pluripotent Stem Cells

The development of induced pluripotent stem cells (iPSC) from somatic cells is possible by their transfection with a combination of transcription factors [4,110,111,112]. One of the approaches for male fertility preservation includes the development of generated iPSCs from somatic cells, such as dermal fibroblasts, blood cells, or keratinocytes, from infertile patients to primordial germ cells (PGCs). These PGCs can either be auto-transplanted into the testes (in vivo) or used in vitro in order to induce the development of gametes [113,114,115,116]. Recently, iPSCs were developed from Klinefelter and NOA patients [117,118]. It was shown that iPSC from rodents, monkeys, and humans can be differentiated into male germ cells [4,85,112,118,119,120,121,122,123,124,125,126,127,128]. The addition of bone morphogenetic proteins (BMPs) to human iPSC induced their differentiation into primordial germ and meiotic cells [129]. Additionally, manipulation of human iPSC with RNA-binding proteins induced the differentiation of their derived germ cells to meiotic stages [112,130]. The addition of conditioned media and retinoic acid to human iPSC cultures led to their differentiation into pre-meiotic, meiotic, and post meiotic (round spermatids) stages [112,131]. It should also be noted that the addition of BMP4 to mouse iPSC cultures led to the development of germ cells. The transplantation of these germ cells into seminiferous tubules of infertile mice induced complete spermatogenesis, including the generation of fertile sperm [125].

Following intra-cytoplasmic sperm injection (ICSI) and transfer of the embryos to recipient females, some of the live offspring developed cancer in the neck and died prematurely [125]. This may indicate that this system may not be safe and requires further research and optimization. The main concern of iPSC use in translational medicine is the transfection of somatic cells with oncogenes, which may lead to the development of tumors in offspring, in addition to the transition of these genes and vectors to the germ line. Additionally, in addition to genetic and epigenetic stability, fertility of the developed human sperm should be clarified (ethical issue). Recently, direct differentiation of human iPSC into pre-meiotic, meiotic, and post meiotic stages, including the development of spermatid-like cells in vitro without genetic manipulation was demonstrated [120]. However, this study needs further validation.

### 3.9. Protection of Spermatogenesis In Vivo from Harmful Gonadotoxic Therapy

Gonadotoxic therapy for male cancer patients may lead to permanent sterility [4,13,14,120]. This could be related not only to the cytotoxic effect on proliferating spermatogonial cells, but also to testicular somatic compartment damage [13,22]. It was demonstrated that gonadotropins and testosterone suppression in rats and mice during or after gonadotoxic treatments induced the development of spermatogenesis (from residual survival endogenous SSCs) and restored fertility [13,22,132,133]. In one human study, hormone suppression after chemo-/radiotherapy preserved spermatogenesis, while, in another study, it did not [133,134]. In addition, hormonal suppression was shown to induce colony formation and the differentiation of transplanted germ cells into the testes of chemo-/radiotherapy treated rats and mice [22,135,136,137,138] and, in some cases, restored their fertility [139,140]. The combination of hormonal suppression (using GnRH-antagonist) and germ cell auto-transplantation or allogeneic transplantation of cryopreserved testicular cells into irradiated monkey testes showed spermatogenesis recovery and a significant increase in sperm counts when compared to irradiated-only testes [54,132]. It is suggested that androgen suppression may lead to efficient effect on the permeability of Sertoli cell barrier, passage of spermatogonial cells through tight junction and the niche of the spermatogonial cells [132,140,141]. It is also possible that androgen suppression increases the homing of spermatogonial cells by decreasing tight junction proteins, and during their proliferation stage by increasing the Wnt5a expression [142].

This system needs to be optimized in order to increase its low efficiency. In addition, further elucidation of the mechanism of action of the hormonal suppression may improve our understanding of the action of the involved hormones in the process of spermatogenesis recovery. The safety issue of using germ cell auto-transplantation still needs to be considered, as mentioned above.

### 3.10. Pharmacological Agents

#### 3.10.1. AS101

The immunomodulator AS101 is a synthetic organic tellurium compound and non-toxic agent [143]. It was shown to increase the survival of mice following chemo-/radiotherapy treatments, indicating an anti-tumor effect, acting synergistically with some chemotherapy in mouse tumor models [144,145]. In humans, a combination of AS101 with chemotherapy significantly reduced the severity of hematopoiesis affected by the chemotherapy [146,147]. Its effect was related to its capacity to decrease the production of anti-inflammatory cytokines, such as interleukin-10, which leads to an increase in GDNF production [148]. The co-treatment of AS101 with cyclophosphamide significantly decreased the harmful effect of cyclophosphamide on the seminiferous tubules in mice and increased the number of offspring compared to the control [149]. The mechanism of its action was suggested to be through decreased DNA fragmentation and an increase in Akt and glycogen synthase kinase-3b (GSK-3b) phosphorylation [149].

To the best of our knowledge, the effect of AS101 has not yet been evaluated in human males in regard to fertility preservation.

#### 3.10.2. Growth Factors/Cytokines

Granulocyte colony-stimulating factor (G-CSF) is a member of the hematopoietic growth factor family, which regulates the proliferation, differentiation, and survival of hematopoietic progenitor cells [150]. Initial studies of the effects of G-CSF on either irradiation- or anticancer drug-induced myelosuppressed animals demonstrated that it can shorten the duration of myelosuppression and increase the absolute numbers of functionally active neutrophils [151,152]. Recently, the radioprotective effects of G-CSF were further investigated with respect to the testicular system. In one study, its administration to adult male mice for three consecutive days before pelvic irradiation protected about 50% of testicular germ cells from radiation induced apoptosis [153]. The study suggested that G-CSF protected from radiation-induced testicular dysfunction via an anti-apoptotic effect that might lead to the recovery of spermatogenesis. In one study, it was reported that treatment of busulfan-injected adult mice with G-CSF for seven days led (10 weeks later) to a significant restoration in spermatogenesis and number of generated sperm in the epididymis as compared to busulfan-injected mice. G-CSF led to increased numbers of PLZF spermatogonia, and G-CSF receptor (Csf3r) was found in undifferentiated spermatogonia. In another study, it was shown that treatment with G-CSF of adult mice before or after busulfan treatment was protective for spermatogenesis for a relatively long time. The G-CSF treatment increased the proliferation of the survival spermatogonial cells [154]. Using a rat model that was treated with G-CSF before busulfan injection, it was demonstrated that the G-CSF-treated animals showed an increase in testis weight and sperm count and viability, along with high expressions of DDX4, DAZL, TP2, PCNA, and BrdU [155]. It was suggested that G-CSF could be used as a possible strategy for future preservation of male fertility in busulfan-treated cancer patients [156].

However, the suggested protocol needs to be optimized and confirmed in human systems. In addition, G-CSF induces granulopoiesis through the induction of bone-marrow stem cell proliferation. Thus, its combination with chemotherapy (cytotoxic to proliferating cells) may lead to a reverse effect.

Both AS101 and G-CSF may affect spermatogenesis also through autophagy. Autophagy is a process in which cytoplasm and organelles of the cells are degraded by proteolytic mechanism. It plays important role in maintaining cellular homeostasis, self-renewal, and differentiation [155,156,157,158,159,160,161]. Autophagy is mediated by mechanistic target of rapamycin (mTOR) signaling pathway [161,162]. mTOR is a key regulator of cell growth, which involves transcription, translation, and autophagy [160,163]. Autophagy is considered to be a crucial process for the formation of important structures and regulation the depletion of other cellular components [161,162]. It was demonstrated that rapamycin (an antiproliferative and immunosuppressant drug) suppressed spermatogenesis (inhibiting spermatogonial cell proliferation that led to sperm reduction) by changing the status of autophagy through inhibiting mTOR-p70S6 kinase signaling pathway in rats [164]. Additionally, AS101 was demonstrated to regulate the phosphorylation of mTOR in mesangial cells [165]. The importance of mTOR in the process of differentiation of mouse myeloid progenitor cells to neutrophil by G-CSF was reported [166]. Therefore, it is possible that some of the functions of AS101 and G-CSF in protection of spermatogenesis might be performed through the regulation mTOR and autophagy process. 

## 4. Conclusions and Remarks

Today, options for treatment of infertile men who cannot generate sperm are not yet available. The technologies in the field that have been proven in animal models are not yet safe for use in humans. On the other hand, different technologies and approaches are being studied in different models including auto-transplantation, hormonal therapy, gene therapy, protection against gonadotoxic agents, pharmacological agents, and in-vitro culture models. The success of these technologies/approaches and their translation to humans offer hope for new strategies to treat male infertility.

## Figures and Tables

**Figure 1 ijms-21-05471-f001:**
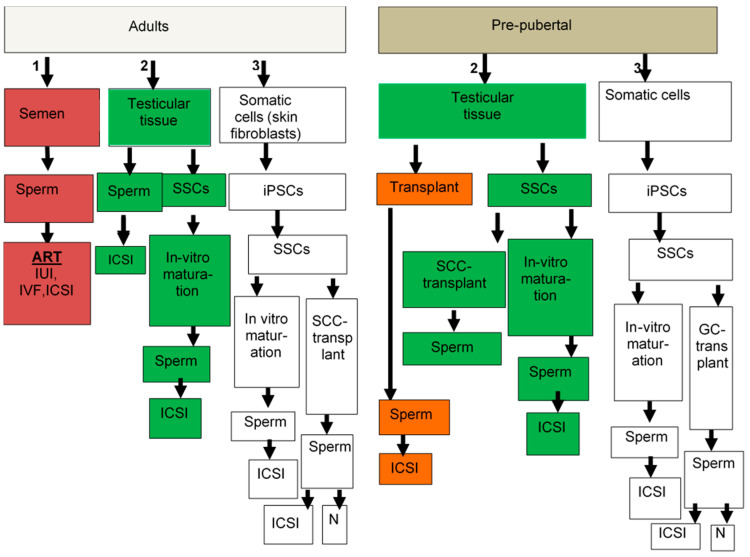
Suggested options for male fertility preservation. Adult patients, if they have sperm in the semen (option #1), can cryopreserve it for ART. If they do not have sperm in their semen, the second option is to use testicular tissue (option #2). If sperm is found in their testicular tissue, it can be used for ICSI. If no sperm is found, then isolated spermatogonia stem cells (SSCs) could be used for in-vitro maturation to generate sperm for use in ICSI. If this technology will not work, or SSCs are not found, then somatic cells could be used to generate iPSC (option #3) and to develop SSCs, which can be used for in-vitro maturation to generate sperm for ICSI or for germ cell transplantation, which leads to generation of sperm in the testes that can normally (N) fertilize oocytes and lead to pregnancy, or if very little sperm is generated, it can be used in ICSI. In pre-pubertal patients, only options #2 and #3 can be used; they do not generate sperm at this age and therefore option #1 is excluded. In pre-pubertal patients, only options #2 and #3 can be used. Testicular biopsies if they contain SSCs (option #2), can be used for transplantation to develop sperm for ICSI, or isolated SSCs can be used either to develop sperm in the testes by germ cell transplantation, or used for in-vitro maturation to develop sperm for ICSI. If testicular tissues do not contain SSCs, then somatic cells (option #3) can be used to generate iPSCs to develop SSCs and used as mentioned above for adult patients.

**Table 1 ijms-21-05471-t001:** Options for fertility preservation and restoration in males.

Approach	Patients	Future Therapeutic Reproductive Strategies	Fertility Options	Advantage	Limitations
Cryopreservation of sperm	Adults	ART ^1–3^	ART	Fertility preservation	(1) Limited quantity and quality of the sperm (2) Not an option for prepubertal patients since they do not yet generate sperm (3) Not recommended after initiation of chemotherapy
Testicular biopsy	Pre-pubertal or Adults	Autologous graft ^4−6^ Autologous germ cell transplantation ^4−6^	Intercourse or ART	Fertility preservation and/or restoration	*For strategies:* (4) Possible contamination with cancer cells * (5) Not yet applicable in human (6) Needs more research for validation, efficiency, and genetic stability (7) Transmissions of mouse virus to human germ line
		Organ culture ^5,6^ In-vitro differentiation culture of SSCs to sperm ^5,6^ Xenograft ^5−7^	ART		
Induced pluripotent stem cells (iPSC)	Adults	In-vitro culture: Differentiation to SSCs ^8,9^	Intercourse or ART	Fertility restoration	(8) Needs more research for validation, safety, efficiency, and genetic stability (9) Not yet applicable in humans
		Differentiation of SSCs to sperm ^8,9^	ART		
Agents to protect spermatogenesis	Pre-pubertal or Adults	Normal fertility ^10−12^ ART ^10−12^	Intercourse or ART	Fertility preservation	(10) Needs more research for possible protective agents (G-CSF, AS101) (11) Needs more research for validation and safety (12) Not yet applicable in humans
Gene therapy **	Adults Or Pubertal	Normal fertility ^13−15^ In-vitro fertilization ^13−15^	Intercourse or ART	Fertility restoration	(13) Needs more research for validation, safety, efficiency, and genetic stability (14) Could be used in testicular somatic cells, but not germ line cells. (15) Not yet applicable in humans

This table summarizes the different options for male fertility preservation and restoration, and the advantages and limitations of these options. ART—Assisted reproductive technology: Intrauterine sperm insemination (IUI), In-vitro fertilization (IVF), Intracytoplasmic sperm injection (CSI). Fertility preservation—preserves fertility that theoretically should be intact (pre-pubertal) or already approved to be present (adults). Fertility restoration—to restore fertility that does not exist. *—It excludes biopsies from non-cancer patients and cancer patients with non-metastatic tumor. **—Could be performed in vivo or in vitro. Uppercase numbers in column “Future therapeutic” indicate the types of limitations. The numbers in column “Future Therapeutic Reproductive Strategies” indicate the type of limitation in the column “limitation”.

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
