# Peer review of "Approaches and Technologies in Male Fertility Preservation"

_ijms, 2020, doi:10.3390/ijms21155471_

Round 1

Reviewer 1 Report

This is a useful manuscript reviewing current and potential options of male fertility preservation in patients at risk of infertility, mainly cancer survivors. These are my concerns/suggestions:

  1. Page 1, line 16 (Abstract): “treatment, surviving rates….”; it is better to say “cancer treatment, surviving rates…”.
  2. Page 1, line 21 (Abstract): What do you mean “approved…”? It is a little misleading.
  3. Page 1, lines 27-28 (Abstract): This is the template! Please write 3 keywords!
  4. Page 1, line 34: “affects Sertoli cells,...”. Please mention the specific function (s) of Sertoli cells that can be affected!
  5. Page 1, line 36: what do you mean “Various signals secreted…”? Either say “Various signal sent…” or “Various substances secreted…”.
  6. Page 1, lines 38-41: what is the main difference between “Adult stem cells” and “Progenitor cells”. Based on the description here, they sound equal as it has been written here. The most important difference between (adult) stem cells and progenitor cells is that stem cells can replicate indefinitely, whereas progenitor cells can divide only a limited number of times. Controversy about the exact definition remains and the concept is still evolving. Please make it clear!
  7. Page 1, lines 43-44: What about Spermatogonia description in “Human”? Although it has been briefly mentioned later in page 2 section 2.4, I recommend to describe it here (with more details) too.
  8. Page 2, line 45: “…compared to spermatogonial cells…”; Which type of Spermatogonial cells?
  9. Page 2, line 51: reference #7 is only about “Leukemia”, please add other references that cover other types of cancer (not cancer treatment) that can disturb spermatogenesis.
  10. Page 2, line 66: “Cells of seminiferous tubules…”; Which type of cells? Please specify it!
  11. Page 2, lines 67-68: “…the type and dose of the drugs…”; Duration of treatment is also important. Please add it!
  12. Page 2, line 78: “….that the interval need…”; what do you mean “interval”? I believe that you are trying to say “….spermatogenesis recovery takes longer if more SSCs have been destroyed”. Correct? Please re-write this part!
  13. Page 2, line 80: “…damage somatic tissue”; what do you mean “somatic tissue”? It is vague.
  14. Page 2, line 81: It should say “…after cytotoxic treatment…”.
  15. Page 2, line 84: “…depletion of the pool of SSCs and seminiferous tubules”; what do you mean “depletion of seminiferous tubules”?
  16. Page 2, line 88: “…quiescent and biologically active SSCs (Adark)”; do you mean “or” instead of “and”?
  17. Page 3, line 94: I do not think the “prevent” is a good term here. These are not preventing options.
  18. Page 3, Table 1 (second part): “For strategis”’ I understand that you are trying to mention limitation of aach future therapeutic reproductive strategy, but I recommend connect them by using asterisks (*, **, ***, ****)
  19. Page 3, Table 1 (forth part): Please mention some examples of protective agents in parenthesis.
  20. Page 4, Figure 1 (option 2 adult): “ICSI” not “ICS”!
  21. Page 4, Figure 1 (option 3 adult): “Somatic cells”; which somatic cells? Skin Fibroblast? Seminiferous tubules somatic cells?
  22. Page 4, Figure 1 (pre-pubertal): Is there “option 1”?
  23. Page 4, Figure 1 (option 2 pre-pubertal): “Transplant” not “Transp”
  24. Page 4, Figure 1 (both adult and pre-pubertal): I recommend using “SSC transplant” instead of “GC-transplant”.
  25. Page 4, lines 126-127: “…ICSI and/or cryopreservation”. It does not make sense! You mean using freshly isolated testicular sperms or freezing them for future ICSI. Please re-write it!
  26. Pages 4 and 5, lines 128-132: The low percentage (10%) of frozen sperm utilization is a different subject than low percentage of sperm cryopreservation prior to cancer treatment.
  27. These two are mixed up here. Please re-write this paragraph!
  28. Page 5, section 3.3: It is important to discuss the low efficiency of SSC transplantation and the importance of SSC in vitro propagation prior to transplantation.
  29. Page 6, line 197: Reference #64 is about “mouse” not “monkey”!
  30. Page 6, line 216: “…in 10% of the seminiferous tubules”; Do you mean “formed seminiferous tubules”?
  31. Page 7, section 3.7: The title of section is misleading as both 2D and 3D culture systems are discussed here.

Author Response

Response to Reviewer 1

I would like to thank the reviewer for the valuable comments and suggestions. Attached please find our response to those comments (red color):

  1. Page 1, line 16 (Abstract): “treatment, surviving rates….”; it is better to say “cancer treatment, surviving rates…”

Done. Line: 15.

  1. Page 1, line 21 (Abstract): What do you mean “approved…”? It is a little misleading.

We changed it to demonstrated.  Line 21.

  1. Page 1, lines 27-28 (Abstract): This is the template! Please write 3 keywords!

Done. Lines: 27-30.

  1. Page 1, line 34: “affects Sertoli cells,...”. Please mention the specific function (s) of Sertoli cells that can be affected!

Done. Lines: 35,36.

  1. Page 1, line 36: what do you mean “Various signals secreted…”? Either say “Various signal sent…” or “Various substances secreted…”.

Corrected. Line 38.

  1. Page 1, lines 38-41: what is the main difference between “Adult stem cells” and “Progenitor cells”. Based on the description here, they sound equal as it has been written here. The most important difference between (adult) stem cells and progenitor cells is that stem cells can replicate indefinitely, whereas progenitor cells can divide only a limited number of times. Controversy about the exact definition remains and the concept is still evolving. Please make it clear!

It corrected. Lines: 40-43.

  1. Page 1, lines 43-44: What about Spermatogonia description in “Human”? Although it has been briefly mentioned later in page 2 section 2.4, I recommend to describe it here (with more details) too.

It was described as recommended. Lines: 47, 48.

  1. Page 2, line 45: “…compared to spermatogonial cells…”; Which type of Spermatogonial cells?

Done. Lines: 49, 50.

  1. Page 2, line 51: reference #7 is only about “Leukemia”, please add other references that cover other types of cancer (not cancer treatment) that can disturb spermatogenesis.

Done. Refs #7,#8.

  1. Page 2, line 66: “Cells of seminiferous tubules…”; Which type of cells? Please specify it!

Done. Line: 71.

  1. Page 2, lines 67-68: “…the type and dose of the drugs…”; Duration of treatment is also important. Please add it!

It was added. Line 73.

  1. Page 2, line 78: “….that the interval need…”; what do you mean “interval”? I believe that you are trying to say “….spermatogenesis recovery takes longer if more SSCs have been destroyed”. Correct? Please re-write this part!

Done. Line 85.

  1. Page 2, line 80: “…damage somatic tissue”; what do you mean “somatic tissue”? It is vague.

       It was changed. Line 86.

  1. Page 2, line 81: It should say “…after cytotoxic treatment…”.

Done. Line 87.

  1. Page 2, line 84: “…depletion of the pool of SSCs and seminiferous tubules”; what do you mean “depletion of seminiferous tubules”?

It was corrected. Lines: 90, 91.

  1. Page 2, line 88: “…quiescent and biologically active SSCs (Adark)”; do you mean “or” instead of “and”?

Corrected. Line 95.

  1. Page 3, line 94: I do not think the “prevent” is a good term here. These are not preventing options.

Corrected. Line 101.

  1. Page 3, Table 1 (second part): “For strategis”’ I understand that you are trying to mention limitation of aach future therapeutic reproductive strategy, but I recommend connect them by using asterisks (*, **, ***, ****)

A change in the table was performed according to the reviewer suggestion.

  1. Page 3, Table 1 (forth part): Please mention some examples of protective agents in parenthesis.

Done.

  1. Page 4, Figure 1 (option 2 adult): “ICSI” not “ICS”!

Corrected.

  1. Page 4, Figure 1 (option 3 adult): “Somatic cells”; which somatic cells? Skin Fibroblast? Seminiferous tubules somatic cells?

Corrected.

  1. Page 4, Figure 1 (pre-pubertal): Is there “option 1”?

There is no option 1, because they do not generate sperm. Addition description was added in the legend of the table. Line 128.

  1. Page 4, Figure 1 (option 2 pre-pubertal): “Transplant” not “Transp”

Corrected.

  1. Page 4, Figure 1 (both adult and pre-pubertal): I recommend using “SSC transplant” instead of “GC-transplant”.

Done.

  1. Page 4, lines 126-127: “…ICSI and/or cryopreservation”. It does not make sense! You mean using freshly isolated testicular sperms or freezing them for future ICSI. Please re-write it!

Done. Lines: 140, 141.

  1. Pages 4 and 5, lines 128-132: The low percentage (10%) of frozen sperm utilization is a different subject than low percentage of sperm cryopreservation prior to cancer treatment.

Corrected. Lines: 142-145.

  1. These two are mixed up here. Please re-write this paragraph!

Done.

  1. Page 5, section 3.3: It is important to discuss the low efficiency of SSC transplantation and the importance of SSC in vitro propagation prior to transplantation.

Done. Lines: 190-200.

  1. Page 6, line 197: Reference #64 is about “mouse” not “monkey”!

Corrected. Ref. # 71.

  1. Page 6, line 216: “…in 10% of the seminiferous tubules”; Do you mean “formed seminiferous tubules”?

Corrected. Line 239.

  1. Page 7, section 3.7: The title of section is misleading as both 2D and 3D culture systems are discussed here.

Corrected. Line 263.

Reviewer 2 Report

The manuscript entitled: Approaches and technologies in male fertility
 preservation, is interesting and well written and updated.

However, it needs improvement as following:

-In line 65, under heading: 2.3. Gonadotoxic agents and male infertility, it is better to add a short sentences related to the gonadotoxic effects of excessive ethanol intake on germ cells and why Sertoli cells are more resistant to gonadotoxic agents. 
-Table 1 should be improved as there accidental lines and written numbers without meaning.

-In Fig.1 what is Transp, transplant?

-In line 259, please explain criteria of sperm-like cells

- in line, 274,  3.7.2.3. D bioprinted scaffold , what is D?

- in lines 301, 302 these sentences are not clear: without similarity to the histology of the testis that did produce testosterone and inhibin B, but did
not produce differentiation of the proliferative spermatogonial cells. Please re-write.

In line 366  under subheading 3.9: Protection of spermatogenesis in vivo from harmful gonadotoxic therapy.  How androgen suppression can restore infertility and maintain spermatogenesis? what is the mechanism?

 In line 371, this sentence; development of spermatogenesis from survival endogenous SSCs, and restored fertility, is not clear.  Please re-write and correct grammar.
- in line 384, and related to pharmacological agents, which may improve fertility, how these agents  do that? is there any role for autophagy or mitophagy as prosurvival mechanisms induced by these agents?

- Can autophagy activation in vivo or in vitro by pharmacological agents help in restoration of fertility and improve sperm quality?

Author Response

Response to Reviewer 2:

I would like to thank the reviewer for the valuable comments and suggestions. Attached please find our response to those comments (red color):

However, it needs improvement as following:

-In line 65, under heading: 2.3. Gonadotoxic agents and male infertility, it is better to add a short sentences related to the gonadotoxic effects of excessive ethanol intake on germ cells and why Sertoli cells are more resistant to gonadotoxic agents.

Done. 

-Table 1 should be improved as there accidental lines and written numbers without meaning.

Done.

-In Fig.1 what is Transp, transplant?

Corrected.

-In line 259, please explain criteria of sperm-like cells.

Done. Lines: 282, 283.

- in line, 274,  3.7.2.3. D bioprinted scaffold , what is D?

Corrected. Line 297.

- in lines 301, 302 these sentences are not clear: without similarity to the histology of the testis that did produce testosterone and inhibin B, but did not produce differentiation of the proliferative spermatogonial cells. Please re-write.

Done. Lines: 324, 325.

In line 366  under subheading 3.9: Protection of spermatogenesis in vivo from harmful gonadotoxic therapy.  How androgen suppression can restore infertility and maintain spermatogenesis? what is the mechanism?

It is explained according to the reviewer comment. Lines: 402-406.

 In line 371, this sentence; development of spermatogenesis from survival endogenous SSCs, and restored fertility, is not clear.  Please re-write and correct grammar. 

Done. Line: 394.

- in line 384, and related to pharmacological agents, which may improve fertility, how these agents  do that? is there any role for autophagy or mitophagy as prosurvival mechanisms induced by these agents?

It is explained according to the reviewer comment. Lines: 450-464.

- Can autophagy activation in vivo or in vitro by pharmacological agents help in restoration of fertility and improve sperm quality?

It is explained according to the reviewer comment.  Lines: 450-464.